# Reinforcement Learning with Adaptive Reward Modeling for Expensive-to-Evaluate Systems

Hongyuan Su [* 1 2]  Yu Zheng [* 1 3]  Yuan Yuan [1]  Yuming Lin [1]  Depeng Jin [1]  Yong Li [1 2]

## Abstract

Training reinforcement learning (RL) agents requires extensive trials and errors, which becomes prohibitively time-consuming in systems with costly reward evaluations. To address this challenge, we propose adaptive reward modeling (AdaReMo) which accelerates RL training by decomposing the complicated reward function into multiple localized fast reward models approximating direct reward evaluation with neural networks. These models dynamically adapt to the agent's evolving policy by fitting the currently explored subspace with the latest trajectories, ensuring accurate reward estimation throughout the entire training process while significantly reducing computational overhead. We empirically show that AdaReMo not only achieves over 1,000 times speedup but also improves the performance by 14.6% over state-of-the-art approaches across three expensive-to-evaluate systems—molecular generation, epidemic control, and spatial planning. Code and data for the project are provided at https://github.com/tsinghua-fib-lab/AdaReMo.

## 1. Introduction

Reinforcement learning (RL) has achieved remarkable success and emerged as the go-to approach for addressing a wide range of tasks (Lowe et al., 2017; Chen et al., 2021; Haarnoja et al., 2018; Farebrother et al., 2024; Yuan et al., 2025). The key to effective RL is a well-defined reward function guiding the agent to update its policy as it navigates the vast solution space (Silver et al., 2021; Levine et al., 2020; Lowrey et al., 2019; Kidambi et al., 2020;

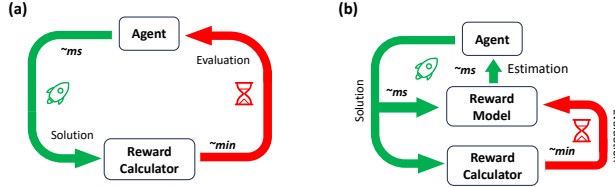

Figure 1. (a) The out-of-sync RL loop for expensive-to-evaluate systems. (b) Our AdaReMo approach adaptively decouples the loop into separate online and offline systems.

Wan et al., 2021). However, despite tasks with immediate and straightforward feedback such as gaming (Mnih et al., 2013; 2015; Vinyals et al., 2019), many real-world tasks involve rewards that are computationally expensive to evaluate (Zhang et al., 2019; Yuan et al., 2023; Yang et al., 2021; Ding et al., 2024; Liu et al., 2024b), creating a loop of fast decision and slow evaluation (Figure 1(a)). For instance, in drug design, reward calculation requires intensive computation of molecular dynamics to determine the absolute binding free energy between a generated molecule and the target (Yang et al., 2021; Lutz et al., 2023), a process that often takes seconds—orders of magnitude slower than the millisecond-scale generation process. Such *expensive-to-evaluate* reward functions create a significant efficiency bottleneck, rendering RL training highly impractical or even infeasible, given the millions of interactions required with the environment.

To mitigate the computational burden of costly reward functions, model-free reinforcement learning (MFRL) often employs proxy reward (Eckmann et al., 2022) or reduced-scale evaluation (Meirom et al., 2021; Liu et al., 2024a). Nevertheless, these approaches typically oversimplify the evaluation, introducing substantial errors into policy optimization and ultimately resulting in suboptimal solutions (Liu et al., 2021). In contrast, model-based reinforcement learning (MBRL) utilizes a world model to approximate the dynamics of the environment, including its reward function (Moerland et al., 2023; Silver et al., 2016; Ha & Schmidhuber, 2018; Lowrey et al., 2019; Wan et al., 2021). Though sidestepping time-consuming reward computation, this approach requires a large amount of high-quality data to train the world model, such as expert demonstrations, which are often sparse in real-world scenarios (Hansen et al., 2023). More importantly, the reward function is notoriously com-

*Equal contribution [1]Department of Electronic Engineering, BNRist, Tsinghua University, Beijing, China [2]Zhongguancun Academy, Beijing, China [3]Massachusetts Institute of Technology, Cambridge, MA USA. Correspondence to: Yu Zheng <yu_zheng@mit.edu>, Yong Li <liyong07@tsinghua.edu.cn>.

*Proceedings of the 42nd International Conference on Machine Learning*, Vancouver, Canada. PMLR 267, 2025. Copyright 2025 by the author(s).

plicated and exhibits drastic local variations depending on the agent's exploration trajectory. Fixed world models struggle to capture these complexities and keep pace with the agent's progress, leading to increasing prediction errors and degraded policy performance as training proceeds (Janner et al., 2019). The computational and modeling challenges significantly hinders the large-scale application of RL in the real world.

In this paper, we propose a general RL approach to decouple the out-of-sync loop of fast decision and slow evaluation into separate online and offline systems. The agent makes rapid decisions in the online system, while the expensive-to-evaluate reward function is offloaded to the offline system (Figure 1(b)). Specifically, we design a neural network-based reward model (RM) to accurately and quickly approximate reward computation, enabling fast interactions with the agent. To address the complexity of the reward function, we introduce adaptive reward modeling (AdaReMo) which approximates rewards only within the agent's currently explored subspace, decomposing the complicated reward function into multiple tractable localized functions. AdaReMo continuously updates the RM using offline data to align with the agent's progress, ensuring low prediction errors and preventing outdated evaluations throughout the training process. With RM synchronizing fast decision and slow evaluation, our approach seamlessly integrates these two systems operating on different timescales, delivering efficient and accurate RL for expensive-to-evaluate systems.

To validate the effectiveness of AdaReMo, we conduct extensive experiments across three challenging real-world scenarios—molecular generation, epidemic control, and spatial planning. All these tasks involve expensive-to-evaluate reward functions, typically requiring 1 to 15 seconds per sample, resulting in prohibitively long training times for convergence with traditional methods. Results show that AdaReMo not only achieves state-of-the-art performance with over 14.6% improvements over existing approaches, but more importantly, it enables highly efficient RL training, delivering a remarkable speedup of over 1,000 times.

The contributions of this paper are summarized as follows,

- We investigate the critical challenge of synchronizing fast decision with slow evaluation, addressing the efficiency bottleneck in RL for expensive-to-evaluate systems.

- We propose adaptive reward modeling which decomposes the complicated and costly reward function into easy-to-capture reward models aligning with the agent's progress, ensuring zero-delay RL training and reliable convergence of decision policies.

- We conduct extensive experiments across three real-world expensive-to-evaluate systems, demonstrating the substan-

tial efficiency advantages and superior decision performance of our approach.

## 2. Related Work

**Decision-making under expensive objectives.** Traditional approaches often rely on heuristic methods or simplified models to reduce computational overhead, yet they sacrifice accuracy and fail to make optimal decisions (Eckmann et al., 2022; Jeon & Kim, 2020). Recent studies have explored the use of surrogate models (Wu et al., 2023; Wang & Van Hoof, 2022) and approximation techniques (Elsayed et al., 2024; Shetty et al., 2024) to replace reward computation while maintaining performance. For instance, bayesian optimization (Balakrishnan et al., 2020; Astudillo & Frazier, 2021) and gaussian processes (Lin et al., 2023; Achituve et al., 2021) have been utilized to efficiently navigate high-dimensional search spaces. Additionally, advancements in parallel computing and distributed systems have scaled up these tasks across multiple processors or GPUs, significantly reducing computation time (Lu et al., 2022).

**Model-based Reinforcement Learning.** Recently, MBRL has shown promise in improving sample efficiency by learning a model of the environment's dynamics (Yu et al., 2021a; Janner et al., 2019; Yuan et al., 2022). Early approaches focused on learning explicit models of state transitions and rewards, facilitating planning and policy optimization without direct interactions with the environment (Silver et al., 2016; Ha & Schmidhuber, 2018; Hafner et al., 2019). Recent advancements have extended MBRL to address real-world complexities, such as high-dimensional state spaces and complex dynamics, through techniques like ensemble methods (Moerland et al., 2023; Wang et al., 2024), uncertainty estimation (Yu et al., 2020; Xu & Liu, 2023), and offline reinforcement learning (Levine et al., 2020; Luo et al., 2023). However, despite these advancements, their limited ability to consistently provide accurate estimates restricts their application to real-world challenging tasks.

## 3. Preliminary

We consider decision-making problems formulated as Markov Decision Processes (MDP) with state space $\mathcal{S}$, action space $\mathcal{A}$, transition probabilities $\mathcal{P}$ and rewards $\mathcal{R} : \mathcal{S} \times \mathcal{A}$ after taking an action in a specific state. The agent learns a policy $\pi_\Theta$ parameterized by $\Theta$, which outputs an action $a = \pi_\Theta(s)$ and receives a reward $r(s, a)$ from the environment. The objective is to maximize the expected return over the entire decision process, formulated as:

$$\max_\Theta E_{\pi_\Theta} \left[ \sum_{t=0}^{\mathcal{T}} \gamma^t r(s_t, a_t) \right] \tag{1}$$

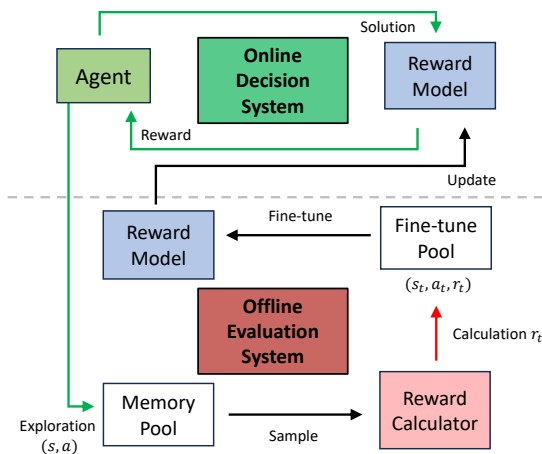

*Figure 2.* The overall framework of AdaReMo, where green and red lines represent fast and slow processes respectively. (Top) In online decision system, the agent updates its policy with real-time feedback from the RM. (Bottom) In offline evaluation system, the RM is continuously finetuned using direct evaluation on the latest exploratory samples of the agent stored in a memory pool.

where $\mathcal{T}$ denotes the time horizon or termination condition and $\gamma$ is the the discount factor. Notably, we focus on expensive-to-evaluate systems where calculating the reward $r(s, a)$ is time-consuming, rendering trivial RL impractical.

## 4. Method

Our method, adaptive reward modeling (AdaReMo) addresses the efficiency bottleneck by (1) disentangling fast and slow processes into two separate systems, (2) synchronizing these systems through a neural network-based reward model (RM) dynamically adapting to the agent's progress.

As illustrated in Figure 2, we begin by offloading heavy reward computations into an offline system, and keep the agent in an online system updating its policy in real-time with the fast RM. We then bridge the speed gap via adaptive updates to the RM using offline data collected by direct reward evaluation, aligning the RM with the agent to ensure accurate reward estimation throughout the entire training process while introducing no computational overhead. Finally, we design synchronous correction, parallel evaluation, and model warm-up to further enhance training efficiency.

### 4.1. Online RL Agent

We develop the RL agent based on graph neural networks (GNN) (Kipf & Welling, 2017) since the graph data structure applies to many real-world tasks, particularly for the three tested scenarios in this work. It is worthwhile to notice that our approach is model-agnostic and can be integrated with other deep learning architectures such as convolutional neural networks (Krizhevsky et al., 2012) and transformers (Vaswani, 2017), which we leave for future work.

We first employ a GNN encoder to transform state observations into dense node and edge embeddings (the graph is defined according to the specific task, see experiments 5), which are calculated as follows:

$$\mathbf{n}_i^0 = \mathbf{W}_n^0 \mathbf{A}_{n_i}, \tag{2}$$

$$\mathbf{n}_i^{l+1} = \mathbf{n}_i^l + \tanh\big(\sum_{j \in \mathcal{N}(i)} \mathbf{W}_n^{l+1} \mathbf{n}_j^l\big), \tag{3}$$

$$\mathbf{e}_{ij}^l = (\mathbf{n}_i^l + \mathbf{n}_j^l)/2 \tag{4}$$

where $\mathbf{A}_{n_i}$ denotes input attributes for nodes, $\mathcal{N}(i)$ denotes the set of neighboring nodes of $n_i$, $\mathbf{W}_n$ is learnable parameters, $l$ denotes GNN layers with a maximum of $L$, $e_{ij}$ represents the edge that connects $n_i$ and $n_j$, and $\mathbf{n}_i$ and $\mathbf{e}_{ij}$ refer to the node and edge embeddings, respectively. The agent then scores each action using these representations with a multi-layer perceptron (MLP), and selects actions based on the probability distribution determined by their scores as follows,

$$s_i = \mathtt{MLP}_p(\mathbf{a}_i), \quad p_i = e^{s_i} / \sum_j e^{s_j}, \tag{5}$$

where $s_i$ and $p_i$ are the score and probability of taking action on node $i$ or edge $i$, and $a_i$ denotes the embeddings of action $a_i$ which indicates node or edge selection.

The agent interacts within the fast online system to collect millions of trajectories for RL training. We optimize its policy $\pi_\Theta$ with PPO (Schulman et al., 2017) in an actor-critic fashion, updating its parameters as follows:

$$\nabla_\Theta J(\Theta) = E\left[\min\left(r_t(\pi_\Theta)\hat{A}_t, \mathrm{clip}(r_t(\pi_\Theta), 1-\epsilon, 1+\epsilon)\hat{A}_t\right)\right]$$
$$\Theta \leftarrow \Theta + \gamma_l \nabla_\Theta J(\Theta), \tag{6}$$

where $\hat{A}^\pi(s, a)$ is the advantage of the state-action pair $(s, a)$ and $\gamma_l$ is the learning rate.

### 4.2. Reward Model

Feeding the direct evaluations back to the agent as rewards for policy optimization is impractical or even infeasible due to the significant computational demands of sophisticated reward functions in expensive-to-evaluate systems. To address this challenge, neural networks offer a promising solution for accelerating evaluation with their robust fitting capabilities and rapid inference speed. Inspired by RLHF (reinforcement learning from human feedback) in finetuning large language model (Touvron et al., 2023), we develop a reward model (RM) for rapid reward estimations. RM replaces intensive computations of the evaluation with a deep neural network to estimate the reward for each state-action pair. We employ another MLP with parameters $\Phi$ for the RM which shares the same GNN encoder with the agent to

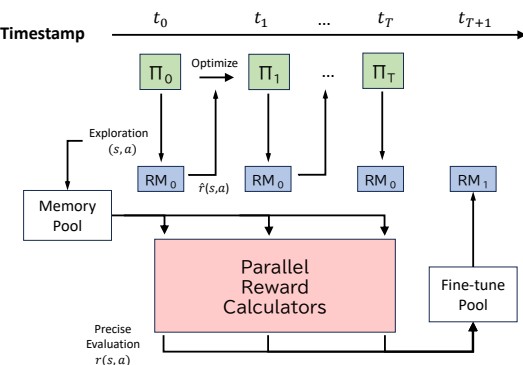

*Figure 3.* The scheme of AdaReMo with asynchronous training framework. Multiple reward calculator constantly evaluate the solution generated by progressively optimized decision policy and for RM fine-tuning. After $T$ policy optimization iterations, RM is fine-tuned to adapt to new decision policy.

significantly reduce training parameters. The RM estimates reward as follows,

$$\mathbf{h}_s = \frac{1}{|V|} \sum_{i \in V} \mathbf{n}_i^L, \quad \hat{r}(s, a) = \mathrm{MLP}_\Phi(\mathbf{h}_s \,\|\, \mathbf{a}), \quad (7)$$

where $V$ denotes the set of nodes, $\mathbf{h}_s$ is the average node representation summarizing the current state, and $\hat{r}(s, a)$ is the estimated reward. The RM is able to provide immediate feedback to the agent, facilitating policy optimization within the online system in real-time.

### 4.3. Adaptive Reward Modeling

While we introduce RM to accelerate RL training, ensuring its accuracy is crucial and requires careful examination. On the one hand, a neural network-based RM may struggle to completely capture the intricacies of the underlying laws of the sophisticated reward function, leading to reward estimates that not always align with direct evaluations. On the other hand, it is impractical to train RM on every possible state $s \in \mathcal{S}$. Consequently, as training proceeds and the agent explores previously unseen or uncommon states, RM may provide erroneous estimates significantly divergent from direct evaluations, which can mislead the agent, hindering its ability to learn the optimal decision policy.

Though RM may not fully capture the complicated reward function, it still has the ability to accurately predict the reward within a reduced and localized state subspace. To ensure accurate reward approximation throughout the entire training process, we propose adaptive reward modeling (AdaReMo) with an asynchronous training framework which updates RM concurrently according to the agent's progress on its policy. The main idea of AdaReMo is to align the definitional domain of RM with the currently explored state subspace by the agent as closely as possible through periodic finetuning. In other words, we decompose

---

**Algorithm 1** Training Process of Online and Offline System

**Online Decision System:**
**Input:** episodelen, policy $\pi_\Theta$, reward model $R_\phi$, transition function $T$, memory pool $M$
**for** $episode = 1$ **to** episodelen **do**
  **for** $t = 1$ **to** $T$ **do**
    $a_t = \pi_\Theta(s_t), \hat{r}_t = R_\phi(s_t, a_t), s_{t+1} = T(s_t, a_t)$
    $M$.push$((s_t, a_t, \hat{r}_t))$
  **end for**
  $G_t = \sum_{k=t}^{T} \gamma^{k-t} \hat{r}_k, \hat{A}_t = G_t - V_\Theta(s_t)$
  $\Theta \leftarrow \Theta + \alpha \hat{A}_t \nabla_\Theta \log \pi_\Theta(s_t, a_t)$
**end for**

**Offline Evaluation System:**
**Input:** memory pool $M$, fine-tune pool $F$, reward calculator $C$
**repeat**
  $M$.pop$((s_t, a_t, \hat{r}_t)), r_t = \mathrm{C}(s_t, a_t)$
  $F$.push$((s_t, a_t, \hat{r}_t, r_t))$
**until** $M$ is $None$

**Adaptive Reward Modeling:**
**Input:** fine-tune pool $F$, reward model $R$, fine-tune interval, fine-tune epoch
**while** $iter \% $ fine-tune interval $== 0$ **do**
  **for** $epoch = 1$ **to** fine-tune epoch **do**
    $F$.randpop$((s_t, a_t, \hat{r}_t, r_t))$
    $L = \sum(\hat{r}_t - r_t)^2$
    $\phi \leftarrow \phi + \nabla_\phi L$
  **end for**
**end while**

---

the full state space $\mathcal{S}$ into multiple subspaces $\{\mathcal{S}_1, \mathcal{S}_2, \ldots, \}$ and fintune the RM using samples within each subspace, thus obtaining $\{\mathrm{RM}_1, \mathrm{RM}_2, \ldots\}$ accordingly.

As illustrated in Figure 3, a fixed-size memory pool is leveraged to store the recent exploratory samples following the first-in-first-out principle, where state-action pairs are sampled to perform direct evaluation for accurate reward $r(s, a)$, which will be added to a fine-tune pool $\mathcal{F}$. As policy optimization iterates, $\mathcal{F}$ is filled with sufficient samples and RM is fine-tuned by the MSE loss as follows,

$$L = \sum_{(s,a) \in \mathcal{F}} (\hat{r}(s, a) - r(s, a))^2. \quad (8)$$

Algorithm 1 shows the training process of our online and offline system with AdaReMo. It is worth noting that the PPO algorithm (Schulman et al., 2017) constrains the magnitude of policy updates to enhance the stability of the agent's learning process. Therefore, each subspace $\mathcal{S}_i$ is compact enough for a neural network $\mathrm{RM}_i$ to accurately approximate, thus guaranteeing consistently low error in reward estima-

**(a)**        **(b)**        **(c)**

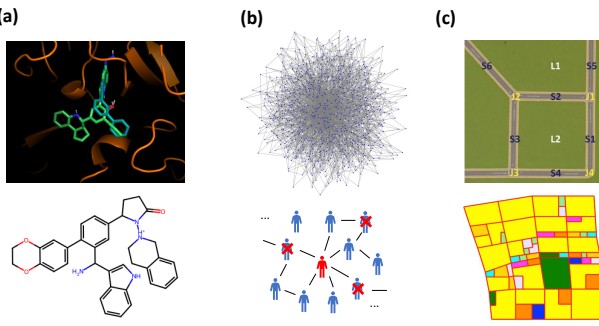

*Figure 4.* Three experimental decision-making tasks, (a) molecular generation, (b) pandemic control and (c) urban spatial planning.

tion and aliging RM with the agent throughout the whole process which we empirically show in experiments.

### 4.4. Training Acceleration

With AdaReMo integrating the decision and evaluation systems operation on different timescales, we introduce synchronous correction and model warm-up to enhance the robustness of reward estimation, as well as parallel computation to further improve the training efficiency.

**Synchronous Correction.** As RM is updated periodically during policy optimization, there can be samples that reside outside the subspace $\mathcal{S}_i$ which become outliers to the definition domain of $RM_i$, leading to errors in reward estimation. To mitigate the impact of such outliers, we introduce a synchronous correction mechanism as follow:

$$\tilde{r}(s,a) = \alpha\hat{r}(s,a) + (1-\alpha)r_c(s,a), \qquad (9)$$

where $r_c(s,a)$ represents the correction term and $\alpha$ is a trade-off parameter. The correction term is obtained from simplified or reduce-scale direct evaluations, allowing for reward rectification without slowing down the training.

**Model Warm-up.** RM's parameters are randomly initialized thus cannot offer reliable reward estimates at the beginning, which can introduce significant noise into or even disrupt policy optimization. Therefore, it is crucial to pretrain the RM before updating the agent's policy. In our implementation, we delay policy optimization until the RM has undergone several fine-tuning intervals, ensuring that the agent consistently receives feedback from a warmed-up RM. While the RM warm-up consumes additional time from the start, it substantially enhances subsequent training efficiency and accelerates model convergence.

**Parallel Computation.** Increasing the number of samples in the fine-tuning pool $\mathcal{F}$ can improve the approximation performance of the RM, as deep learning often benefits from more training data. Therefore, we employ parallel computation to significantly augment the dataset during fine-tuning intervals. Specifically, we implement simultaneous calculation of $K$ reward calculators using multi-threaded

*Table 1.* Time spent on evaluating molecular generation (MG), pandemic control (PC) and urban spatial planning (USP).

| Evaluation | MG | PC | USP |
|---|---|---|---|
| RM | 0.02 | 0.02 | 0.01 |
| Simplified | 0.03s | 0.12s | 0.01s |
| Precise | 8.6s | 15.2s | 10.5s |

programming, and the hyper-parameter $K$ is determined by computational resources. Each calculator pulls the latest solution from the memory pool, completes its reward evaluation, and then initiates another round of calculation.

## 5. Experiments

We investigate three challenging tasks with expensive-to-evaluate reward functions: molecular generation, pandemic control, and urban spatial planning, as shown in Figure 4. Additionally, we summarize the time spent on the corresponding evaluations in Table 1, where using the RM results in a speedup of approximately 1,000 times compared to employing the precise reward functions.

### 5.1. Molecular Generation

Molecular generation aims to identify novel molecules that bind most effectively to protein targets, where RL has become a promising method due to its ability in searching a vast solution space (Yang et al., 2021; Lutz et al., 2023). A molecule can be represented as a graph $G = (V, E)$, where atoms are nodes and bonds are edges. The generation process is equivalent to graph expansion, where the agent's action corresponds to adding a new fragment (a set of nodes) connected by a bond (edge) to the existing molecular structure at a specified attachment site.

Evaluating the quality of the molecule presents significant challenges. The molecular docking program is widely used to provide precise measurements of the therapeutic potential of molecules. Through computationally expensive molecular dynamics-based simulations, the program calculates accurate binding free energy to identify hit compounds. Here, we employ AutoDock Vina (Trott & Olson, 2010; Eberhardt et al., 2021), a docking engine with outstanding accuracy, to evaluate the effectiveness of the generated molecules in targeting proteins, as the reward in the MDP. To compare the performance of each approach, we investigate the average score of the top 5%-scored generated molecules, and hit ratio which is the proportion of docking scores exceeding reference thresholds. Notably, the molecular docking program requires several seconds to obtain precise results for a single sample, which is unaffordable for traditional RL methods.

*Table 2.* Performance comparison on molecular generation with respect to *Top 5% Score* (T5) and *Hit Ratio* (HR).

| Method | FA7 | | PARP1 | | 5HT1B | |
|---|---|---|---|---|---|---|
| | T5↑ | HR↑ | T5↑ | HR↑ | T5↑ | HR↑ |
| HierVAE | $9.4 \pm 0.1$ | $0.06 \pm 0.01$ | $12.2 \pm 0.1$ | $0.25 \pm 0.01$ | $11.9 \pm 0.1$ | $0.12 \pm 0.01$ |
| LIMO | $9.8 \pm 0.7$ | $0.11 \pm 0.02$ | $11.9 \pm 1.0$ | $0.18 \pm 0.04$ | $10.3 \pm 0.8$ | $0.20 \pm 0.04$ |
| MolDQN | $8.2 \pm 0.3$ | $0.02 \pm 0.01$ | $10.5 \pm 0.2$ | $0.04 \pm 0.02$ | $9.8 \pm 0.1$ | $0.11 \pm 0.01$ |
| FREED | $\underline{10.1} \pm 0.2$ | $\underline{0.23} \pm 0.04$ | $\underline{12.8} \pm 0.3$ | $\underline{0.35} \pm 0.09$ | $12.2 \pm 0.2$ | $\underline{0.41} \pm 0.10$ |
| MBPO | $9.7 \pm 1.1$ | $0.18 \pm 0.07$ | $11.6 \pm 0.9$ | $0.26 \pm 0.08$ | $12.3 \pm 0.5$ | $0.33 \pm 0.08$ |
| GRPO | $9.6 \pm 0.2$ | $0.21 \pm 0.03$ | $11.3 \pm 0.5$ | $0.22 \pm 0.06$ | $11.8 \pm 0.5$ | $0.38 \pm 0.08$ |
| RLOO | $10.0 \pm 0.4$ | $0.22 \pm 0.02$ | $12.0 \pm 0.6$ | $0.29 \pm 0.06$ | $\underline{12.5} \pm 0.4$ | $0.40 \pm 0.08$ |
| Ours | $\mathbf{10.4} \pm 0.2$ | $\mathbf{0.29} \pm 0.03$ | $\mathbf{13.1} \pm 0.4$ | $\mathbf{0.42} \pm 0.05$ | $\mathbf{12.7} \pm 0.4$ | $\mathbf{0.48} \pm 0.09$ |
| impr% | +1.9% | +19.4% | +1.6% | +20.0% | +3.2% | +17.1% |

We experiment on three protein targets, FA7, PARP1, and 5HT1B, which are commonly studied in medical research (Yang et al., 2021; Nautiyal et al., 2015). We compare our method with state-of-the-art molecular generation baselines, which are (1) non-RL algorithms including HierVAE (Jin et al., 2020) and LIMO (Eckmann et al., 2022), (2) MFRL-based methods including MolDQN (Zhou et al., 2019) and FREED (Yang et al., 2021), (3) MBRL-based algorithm MBPO (Janner et al., 2019), GRPO (Shao et al., 2024) and RLOO (Ahmadian et al., 2024). For each metric, we repeat experiments with 10 different random seeds and record the mean and the standard deviation.

Table 2 illustrates the performance of different approaches, which confirm the necessity of precise evaluation. Specifically, using the simplified evaluation, MolDQN consistently generates molecules with poor binding quality, showing an sharp performance decline of 21.9% and 93.5% in *Top 5% Score* and *Hit Ratio*, respectively. In contrast, benefit from the dynamic-based precise evaluation, our approach and FREED always achieve the optimal and suboptimal performance, with an average improvement of 13.2% and 10.9% over other RL methods.

Our method has remarkable advantages over other baselines. For all the three different target proteins, our method demonstrates the best generation quality. Specifically, the *Hit Ratio* of our methods improves other MFRL methods by over 8.3%. Meanwhile, by employing AdaReMo, the agent is able to capture the localized reward function accurately, reaching an improvement of 6.3% over MBRL approach.

## 5.2. Epidemic Control

Mitigating the impact of a pandemic requires strategic allocation of limited resources such as quarantine facilities and vaccine supplies within social networks. The challenge in epidemic control stems not only from the vast and intricate social networks but also from the difficulty in accurately modeling disease dynamics. Pandemic control can be conceptualized as sequentially selecting nodes on a social network $G = (V, E)$ to be temporarily isolated, where

*Table 3.* Pandemic control performance measured by *Healthy* (H) and *Contained* (C).

| Method | CA-GrQc | | SNAP | |
|---|---|---|---|---|
| | H↑ | C↑ | H↑ | C↑ |
| HSB | $31.7_{\pm 1.8}$ | $8.8_{\pm 0.2}$ | $22.3_{\pm 1.1}$ | $3.0_{\pm 0.3}$ |
| KED | $30.2_{\pm 2.2}$ | $8.7_{\pm 0.7}$ | $22.1_{\pm 2.4}$ | $2.3_{\pm 0.4}$ |
| GBP | $32.9_{\pm 0.5}$ | $9.1_{\pm 0.3}$ | $23.8_{\pm 0.4}$ | $3.1_{\pm 0.1}$ |
| RLGN | $\underline{35.8}_{\pm 2.3}$ | $\underline{9.8}_{\pm 1.4}$ | $\underline{25.6}_{\pm 3.1}$ | $\underline{3.4}_{\pm 1.1}$ |
| MBPO | $27.2_{\pm 6.6}$ | $7.3_{\pm 3.5}$ | $15.4_{\pm 6.0}$ | $1.3_{\pm 0.6}$ |
| Ours | $\mathbf{39.8}_{\pm 3.4}$ | $\mathbf{10.4}_{\pm 1.2}$ | $\mathbf{28.2}_{\pm 4.8}$ | $\mathbf{3.8}_{\pm 0.9}$ |
| impr% | +7.4% | +7.2% | +10.2% | +11.5% |

$V$ represents individuals and $E$ denotes their interpersonal contacts (Meirom et al., 2021).

For accurate evaluation of epidemic control, it is common to employ susceptible-infectious-recovered (SIR) model (Kermack & McKendrick, 1927) to capture the propagation dynamics of pandemic. By predicting the health status of individuals at each stage of propagation multiple times, SIR simulation provides a more comprehensive decision assessment. Performance of epidemic control is measured by *Healthy* representing the final proportion of healthy individuals, and *Contained* indicating the proportion of simulations where *Healthy* exceeds 60% (Meirom et al., 2021).

We utilize large-scale real-world contact networks CA-GrQc (Rossi & Ahmed, 2015) and SNAP (Leskovec & Krevl, 2014), which are extensively studied in epidemiological research. Consistent with prior studies (Meirom et al., 2021), the agent isolates 2% of the total population, and we simulates 25 propagation steps with SIR model using 20 different seeds (3 for synchronous correction). Additionally, the parameters of the SIR model are set with an infectious rate $\beta = 0.08$ and a recovery rate $\gamma = 0.2$, informed by real-world pandemic propagation (Yu et al., 2021b). We compare our method against (1) classic approaches including KED (Tong et al., 2012), GBP (Kimura et al., 2008) and heuristic search approaches based on betweenness (HSB), (2) MFRL approach RLGN (Meirom et al., 2021), and (3) MBRL approach MBPO (Janner et al., 2019).

*Table 4.* The basic attributes of the real-world communities.

| Community | Location | Area | Round | Grids |
|-----------|----------|------|-------|-------|
| HLG | Beijing | $3.67km^2$ | $7.7km$ | 38 |
| DHM | Beijing | $3.35km^2$ | $8.0km$ | 48 |
| HZ | Guangzhou | $2.96km^2$ | $6.9km$ | 54 |

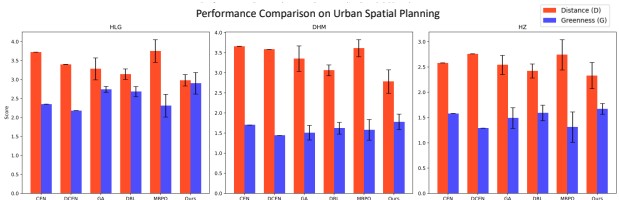

*Figure 5.* Urban spatial planning performance measured by Distance (D, the lower the better) and Greenness (G, the higher the better).

Table 3 illustrates the results of our approach and baselines. First, classic algorithms cannot address complex epidemic control tasks well. HSB heavily relies on manually designed rules and often fails to capture the essential characteristics of epidemics, resulting in poor performance across networks and metrics. Additionally, KED and GBP also show noticeable declines compared to RL approaches. Specifically, the risk of infection increases by at least 11.7%, while the probability of containing outbreaks decreases by at least 17.2%. Second, our approach achieves the best performance on both *Healthy* and *Contained* across different social network, demonstrating its outstanding ability in pandemic control. Specifically, our generated control strategy prevents 3.3% of the population from infection and is the only strategy with a containment ratio higher than 10% among all the methods. Moreover, our approach improves by 45% over the MBRL approach due to the adaptability of AdaReMo.

### 5.3. Urban Spatial Planning

Rationalizing the functional division of limited urban land presents a challenge, requiring consideration not only of the city's actual development needs but also of the functional interconnections between different types of land uses. The task of urban spatial planning can be formulated as sequentially selecting edges on a city graph $G = (V, E)$, where $V$ represents community lands and road segments, and $\mathcal{E}$ indicates their spatial adjacency (Zheng et al., 2023).

To incorporate realistic human mobility into community layout evaluation, we utilizes the state-of-the-art urban mobility simulation method SAND (Yuan et al., 2023) for evaluation. With thousands of reconstructed daily trajectories of residents, each layout for community is measured by *Distance* and *Greenness*. *Distance* indicates the accessibility of the community layout, quantified by the average daily trajectory length per resident. *Greenness* signifies the residents' exposure to green spaces, calculated by the average daily

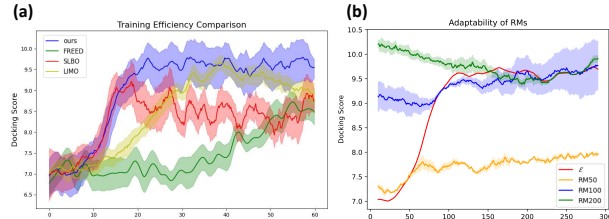

*Figure 6.* (a) Training efficiency comparison of four RL-based methods in molecular generation. (b) The adaptability of RMs, where solutions are evaluated at each policy optimization iteration. Best viewed in color.

passes through green areas per resident.

We experiment on three real-world communities in China (Zheng et al., 2023), initially bordered by main roads and designated residential areas, with further details provided in Table 4. The task involves partitioning the original community layouts to allocate areas for green spaces, businesses, offices, schools, hospitals, recreation, and residential purposes and ends when all the requirements are satisfied. In the implementation, a complete simulation contains the weekly trajectories of 1,000 virtual residents, while in synchronous correction, we decrease to 100 residents. For the performance comparison, we include (1) traditional planning concepts such as centralized (CEN) and decentralized (DCEN), (2) genetic algorithm (GA) (Zheng et al., 2023), (3) MFRL-based method DRL (Zheng et al., 2023) and (4) MBRL approach MBPO (Janner et al., 2019).

Figure 5 illustrates the results of our approach and baselines. While MFRL effectively explores the solution space, the imprecise rewards received by the agent limit the efficacy of layout planning. By employing a realistic mobility-based evaluation, our method significantly improves community layouts, reducing average travel distance from 3.06 to 2.78 (-9.2%) and increasing average green space visits from 1.62 to 1.78 (+9.9%). Additionally, within three RL-based approaches, MBRL approach consistently produces layouts with poor accessibility and green space exposure, showing an average performance decline of 18.8% and 14.2% in *Distance* and *Greenness*, respectively. Notably, although our method achieves the best planning outcomes overall, MBPO performs the worst among all baselines, underscoring the critical role of the AdaReMo.

### 5.4. Effectiveness and Efficiency of AdaReMo

RM plays a pivotal role as the key bridge between online optimization and offline evaluation in our approach. To illustrate the difference on training efficiency between DRL-based methods, we present the evaluation metrics of the generated solutions after the same optimization time in Figure 6(a). While RL using direct evaluation (FREED) initially shows a slight advantage, the performance of our

**(a)** 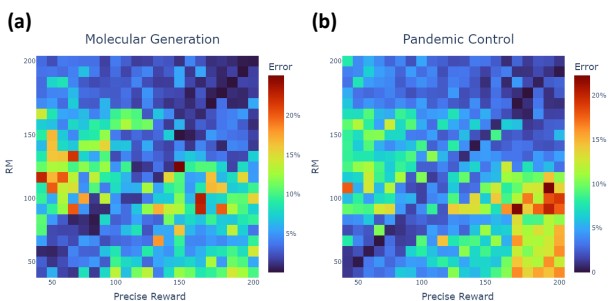

**(b)** 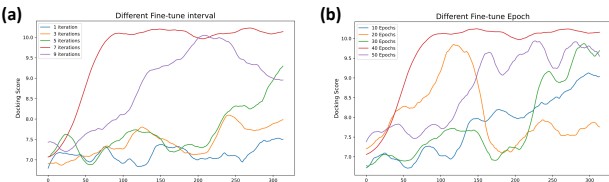

*Figure 7.* The visualization of RMs' estimation errors in (a) molecular generation and (b) pandemic control. The element at $(i, j)$ represents the error in applying the RM at $j$ iteration to estimate the solution at $i$ iteration.

*Figure 8.* The effectiveness of AdaReMo with different hyperparameters. (a) Impact of varying fine-tune intervals on training efficiency and convergence. (b) Influence of varying fine-tune epochs on metric performance. Best viewed in color.

approach quickly catches up and surpasses that of FREED, with an increase of more than 17.6%. This observation can be attributed to the insufficient fine-tuning of RM to capture the complex dynamics of the environment during the early training stages, which introduces reward estimation errors. Benefiting from accurate evaluations and fast RM estimations, our approach efficiently optimizes the policy and achieves the fastest convergence to generate quality molecules. In contrast, even after several days of training, the performance of RL without RM improved by only a slight 7.1%, highlighting the necessity of the RM which sidesteps the heavy computational burden of reward calculation with fast estimations using neural networks.

In addition to improving overall training efficiency, the adaptability of the model to a continuously optimized decision policy is also examined. We randomly select solutions with accurate evaluations from the fine-tuning pool $\mathcal{F}$ during the training process. Fine-tuned RMs are saved at iterations 100, 300, and 500 (denoted as RM100, RM300, and RM500, respectively) to estimate molecular properties. Figure 6(b) presents the evaluations of the molecules alongside the corresponding RM estimations. Specifically, all RMs accurately estimate molecules generated around their respective iterations, with an average relative loss of less than 0.8% and a relative standard deviation of less than 2.1%. In contrast, RMs exhibit estimations with a relative loss exceeding 37.5% and extremely high variance for molecules outside their trained state subspace.

To further visualize the consistency of between the RM and the agent, we calculate the errors between RM estimations and precise evaluation during the training process. As shown in Figure 7, the elements on the diagonal are approximately 0, indicating that the RM can always adapt to the current solutions with accurate estimations. When the iteration differs, the error of estimation rises significantly. Notably, two areas of high variance can be observed: the lower right corner and the middle left. The former is due to the RM's initial inability to effectively learn the dynamics of the environment, while the latter indicates a sharp shift

in the distribution of the reward function. These findings underscore the necessity of AdaReMo and the asynchronous training framework, ensuring that RM remains aligned with the decision policy.

### 5.5. Empirical Analysis of AdaReMo

Choosing the optimal timing for fine-tuning RM is crucial for enhancing the efficiency of AdaReMo. Here, two critical hyper-parameters for the asynchronous training framework are investigated: fine-tune interval and fine-tune epoch.

We first explored different fine-tuning intervals, ranging from 1 to 9 iterations, and trained the RM accordingly. As shown in Figure 8(a), a short fine-tuning interval (e.g., 1 iteration) impedes effective RM updates, leading to policy optimization failure. Specifically, optimization efficiency drops to less than 75% of default settings, and the convergence fails even after 300 iterations. Conversely, excessive sampling also hinders agent learning efficiency, resulting in a 5.7% decrease in performance. With a short fine-tuning period, the data in the fine-tuning pool $\mathcal{F}$ closely track the exploration subspace but may lack sufficient samples due to time-consuming evaluation. On the other hand, a longer fine-tuning interval makes $\mathcal{F}$ denser and more efficient for RM fine-tuning, yet risks lagging RM updates significantly behind policy optimization, leading to misalignment between the RM and the agent.

Similarly, we varied the number of finetuning epochs to examine its influence, as depicted in Figure 8(b). Both excessive and insufficient epochs resulted in significant performance decrease of 7.5% and 15.1%, respectively. Fewer epochs enable quicker RM feedback to the agent but may compromise fine-tuning effectiveness. Conversely, more epochs facilitate thorough RM adaptation but require additional time and risk overfitting. The optimal solution was found with 40 fine-tuning epochs, matching the duration of a single policy iteration and demonstrating efficient time utilization. Adequate sampling in $\mathcal{F}$ during this period ensures effective RM fine-tuning, allowing agents to receive accurate feedback in subsequent iterations with updated RM. Insufficient time may diminish fine-tuning quality, while excessive time may force agents to iterate multiple times

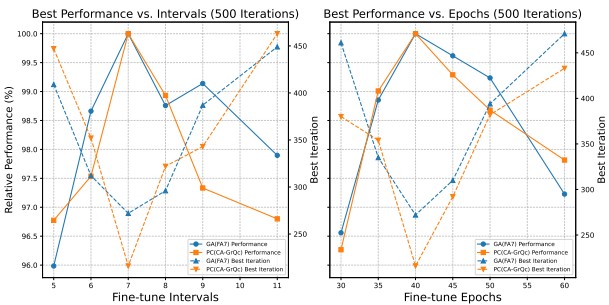

*Figure 9.* Best Performance and corresponding iterations of our framework with different hyper-parameters.



*Figure 10.* Results of the ablation studies on the molecular generation task, where synchronous correction, model warm-up, and parallel computation components are removed individually.

with outdated RM, wasting exploration efforts.

It is important to clarify that above discussions do not imply that our framework is highly sensitive to hyper-parameters. We extended the number of optimization iterations from 300 to 500 and recorded the model's best performance along with the corresponding iteration number across various hyper-parameter settings. The results are presented in Figure 9. Under suboptimal hyperparameter configurations, training efficiency is substantially reduced, as the number of iterations required to reach peak performance increases from 216 to 463, resulting in more than twice the optimization time. Nevertheless, the final performance of the model remains remarkably stable, with less than a 4% difference compared to the optimal one. These results demonstrate that while appropriate hyper-parameter selection can significantly improve training efficiency, the framework itself is robust and maintains competitive performance even in the presence of suboptimal hyper-parameters.

Additionally, we would like to provide practical guidelines that enable users to efficiently estimate an approximate range for the optimal hyper-parameters, ensuring the method's reliability and reproducibility without exhaustive tuning. First, the optimal number of fine-tuning interval can be approximated with the complexity of the reward function, positively correlated with its evaluation time. For example, Table 1 indicates that reward computation in pandemic control is more time-consuming than molecular design, and grid search identified an optimal fine-tuning interval of 9 and 7 (see Figure 8(a)) for the two scenarios, respectively, suggesting that a more complex reward function requires additional samples for effective RM fine-tuning. Second, the optimal number of fine-tuning epoch should balance the fine-tuning duration with the policy optimization. Specifically, we can measure the time $t_1$ required for one sampling and policy optimization iteration. Then, based on the previously determined fine-tuning interval and the reward function's computation time, we can estimate the number of samples needed for fine-tuning and compute the time $t_2$ per fine-tuning epoch. The theoretically optimal number of epochs is $N \approx \frac{t_1}{t_2}$, providing a robust starting point. Search within a

small neighborhood of this estimated value typically yields the optimal setting, as validated in our experiments.

### 5.6. Ablation Study

We conduct a series of ablation studies to evaluate the individual contributions of key components within our proposed framework. The performance of each variant which remove the corresponding component are presented in Figure 10.

The exclusion of any single component leads to a measurable degradation in performance. In particular, the removal of synchronous correction introduces heightened stochasticity and systematic bias, resulting in an average performance decline of approximately 17%, with the most severe degradation reaching 45.2%. Meanwhile, excluding the model warm-up phase yielded a consistent performance drop of around 7% across evaluation metrics. This degradation can be attributed to the policy network being initially exposed to randomly generated rewards, which injects noise into the training process and impairs early-stage optimization. Likewise, the removal of parallel computation capabilities diminished training efficiency by restricting the reward model's capacity to process a sufficient volume of samples for accurate fine-tuning. This constraint led to an estimated 10% reduction in overall performance.

### 6. Conclusion

This work proposes AdaReMo, a general and efficient reinforcement learning approach for systems involving costly reward functions. We introduce a reward model to approximate reward calculation, which disentangles the fast decision and slow evaluation into distinct online and offline systems, enabling efficient policy training without any delays. Meanwhile, the reward model continuously adapts to the agent's progress, ensuring accurate reward approximation throughout the entire training process. AdaReMo displays competitive performance in molecular generation, epidemic control, and spatial planning. Looking ahead, we plan to further explore the universality of our approach across different deep learning architectures besides GNN, as well as broader expensive-to-evaluate tasks.

## Acknowledgements

This work is supported in part by National Natural Science Foundation of China under grant U23B2030 and Zhongguancun Academy Project No.20240303. This work is also supported in part by Tsinghua University-Toyota Research Center and Beijing National Research Center for Information Science and Technology (BNRist).

## Impact Statement

This paper presents work whose goal is to advance the field of Machine Learning. There are many potential societal consequences of our work, none which we feel must be specifically highlighted here.

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
