# OpenReview forum: "Reinforcement Learning with Adaptive Reward Modeling for Expensive-to-Evaluate Systems"
_ICML.cc/2025/Conference — ICML 2025 poster_

### Official Review · Reviewer_joeE · 2025-03-14

**Overall Recommendation:** 4

**Summary:**

The paper presents AdaReMo, an approach designed to accelerate reinforcement learning (RL) in systems where reward evaluations are computationally expensive. The key idea is to decouple the RL loop—where decisions are made quickly—from the reward evaluation process that is slow and costly. AdaReMo achieves this by introducing a neural network–based reward model (RM) that approximates the true reward function. To handle the complexity and variability of real-world reward functions, the approach adaptively decomposes the overall reward function into multiple localized reward models that are trained on the agent’s most recent exploratory data. This adaptive reward modeling is integrated into an asynchronous training framework where the online decision system rapidly collects trajectories using fast RM predictions, while an offline evaluation system periodically updates the RM using precise but expensive reward calculations.

**Claims And Evidence:**

The paper claims that AdaReMo can decouple the fast decision-making loop from the slow, expensive reward evaluations, achieving over 1,000× speedup and about 14.6% performance improvement across three distinct real-world tasks. The extensive experiments in molecular generation, epidemic control, and urban spatial planning provide strong empirical support for these claims. However, one might ask whether the reported gains depend heavily on specific task formulations or the particular settings (e.g., the chosen hyperparameters and the design of the reward model) used in the experiments. Are the improvements robust across a wider variety of expensive-to-evaluate systems?

**Essential References Not Discussed:**

N/A

**Experimental Designs Or Analyses:**

The experiments span three different domains, each chosen to represent a class of expensive-to-evaluate tasks. The paper performs ablation studies on key hyperparameters (e.g., fine-tuning interval and epochs), which helps to understand the sensitivity of the adaptive reward model.

**Methods And Evaluation Criteria:**

The idea of decoupling the online decision system from the offline evaluation system is a natural fit for scenarios where reward computations are very expensive. Using an adaptive neural reward model that fine-tunes based on recent exploratory data is an inventive solution to keep the online loop fast. The evaluation criteria—such as Top 5% Score and Hit Ratio for molecular generation, Healthy and Contained metrics for epidemic control, and accessibility/greenness metrics for spatial planning—are well chosen to reflect the quality of decisions under high computational cost.

**Other Comments Or Suggestions:**

No comments

**Other Strengths And Weaknesses:**

- The decoupling of the RL loop into online and offline components using an adaptive reward model is an interesting solution to a long-standing computational bottleneck in RL.
- By demonstrating significant speedup and performance gains on challenging real-world tasks, the paper makes a strong case for its practical relevance.
- The paper is clear in describing the architecture, algorithm (with pseudocode in Algorithm 1), and the adaptive training framework. Detailed experimental results and visualizations (e.g., Figures 3–8) further aid understanding.

**Questions For Authors:**

No questions

**Relation To Broader Scientific Literature:**

The paper builds on ideas from surrogate modeling and model-based RL, and it resonates with recent work in reinforcement learning from human feedback (RLHF) by adapting techniques to approximate expensive evaluations.

**Theoretical Claims:**

The paper primarily emphasizes an empirical demonstration of its ideas rather than formal theoretical proofs. The conceptual justification for adaptive reward modeling is well articulated, yet there isn’t a formal mathematical proof guaranteeing, for instance, convergence or error bounds for the reward model approximation.

---

> ### Author Rebuttal · Authors · 2025-04-01
>
> Dear Reviewer joeE,
>
> Thank you for your thoughtful and constructive feedback. We appreciate your recognition of the core idea behind AdaReMo, particularly the decoupling of the fast decision-making loop from the slow, expensive reward evaluations. We hope the following responses can address your concerns.
>
> **Q1:** Hyperparameters settings.
>
> **A1:** Thanks for your constructive comments.
> Besides the grid search results in Figure 8, we would like to provide **practical guidelines** that enable users to efficiently estimate an **approximate range for the optimal hyperparameters**, ensuring the method’s reliability and reproducibility without exhaustive tuning.
>
> First, the optimal number of **fine-tuning interval** can be approximated based on the complexity of the reward function, which is **positively correlated with its evaluation time**. For example, Table 2 indicates that reward computation in pandemic control is **more time-consuming** than molecular design, and grid search identified an optimal fine-tuning interval of 9 and 7 (Figure 8a) for the two scenarios, respectively, suggesting that a more complex reward function requires additional samples for effective RM fine-tuning.
>
>
> Second, the optimal number of **fine-tuning epoch** should balance the fine-tuning duration with the policy optimization (Section 5.5). Specifically, we can measure the time $t_1$ required for one policy optimization iteration. Then, based on the previously determined fine-tuning interval and the reward function’s computation time, we can estimate the number of samples needed for fine-tuning and compute the time $t_2$ per fine-tuning epoch. **The theoretically optimal number of epochs is $N \approx \frac {t_1}{t_2}$**, providing a robust starting point. Search within a small neighborhood of this estimated value typically yields the optimal setting, as validated in our experiments.
>
> We sincerely appreciate your constructive comment, and we have added the above guidelines on hyperparameters in Section 4.5 of the revised manuscript.
>
>
> **Q2:** Are the improvements robust across a wider variety of expensive-to-evaluate systems?
>
> **A2:** Thank you for your insightful question. The proposed AdaReMo is designed as a general solution for expensive-to-evaluate systems. To demonstrate its versatility, we conducted extensive experiments across three quite diverse scenarios: molecular design, pandemic control, and urban planning. The choice of tasks has also been acknowledged by Reviewer kWsg.
> We sincerely appreciate your suggestion and consider the inclusion of a broader range of expensive-to-evaluate tasks an important direction for extending our approach. In particular, we plan to investigate the effectiveness of AdaReMo in additional domains such as robotics and autonomous driving.
>
> Thank you once again for your valuable comments, please let us know if you have additional questions.

---

### Official Review · Reviewer_kWsg · 2025-03-14

**Overall Recommendation:** 3

**Summary:**

This work deals with the RL setting in which we have access to a
reasonably fast simulator of the system, but the reward evaluations
are slow. They employ the idea of modeling the reward to enable taking
advantage of the fast simulation to update the policy.  In particular,
they use an asynchronous scheme where they are running multiple true
reward evaluations in parallel that are added to a buffer for training
the reward model. The states for reward evaluation are sampled from a
buffer of recent states encountered in policy optimization.  They
optimize the policy using PPO and their trained reward model.  The
training process also employs a warmup phase for initial reward model
training, as in the beginning the reward model is inaccurate and can
potentially lead to detrimental policy learning.  Another part of
their proposal is adding a correction term to the reward during policy
optimization; in particular, even though computing the precise reward
is slow, there exists a faster surrogate $r_c$ for the tasks that they
consider. As their learned reward model may be wrong, they blend this
surrogate into their reward prediction to aid with dealing with
outlier data $\alpha r + (1-\alpha) r_c$.  They consider the tasks of
molecular generation, pandemic control and urban spatial planning. As
all of these tasks involve a graph-based structure, they employ Graph
Neural Networks for the policy.  The performance improved across
several baselines (by around ~10%) including classic approaches as
well as the model-based RL algorithm MBPO. The ablation study showed
that the method is sensitive to the number of fine-tuning intervals
and epochs for training the reward model.


-------------------------------------------------------
Update:
Thank you for the response; this addresses most of my concerns, so I increased the score.

One point that still caught my eye was that the results you currently reported were based on the value at the best iteration; however, in the paper some of the learning curves were quite erratic, oscillating up and down, so even if the best iteration result is good, the learning may still be unstable (at least from the presented result, I can't rule out this possibility). However, perhaps in terms of your application, you are mainly interested in the best result, so I can also see it being argued the other way.

**Claims And Evidence:**

There are many claims about improved performance, and they also introduce
methods such as blending the reward value estimates, the GNN model, etc.

Currently, I am not convinced by the claims as the number of seeds is
either low, or not specified. For the molecular task, they said they
repeated the experiment with 3 seeds, which is typically considered
very low (e.g., see https://arxiv.org/abs/2108.13264 Deep
Reinforcement Learning at the Edge of the Statistical Precipice).

Moreover, some ablations are missing, e.g., an ablation for the
blended reward in equation 9 is missing.

A major concern for me is the sensitivity study in Figure 8 about the
number of fine-tuning intervals (tested in [1, 3, 5, 7, 9]) and the
number of epochs (tested in [10, 20, 30, 40, 50]). The method only
provides reasonable performance for the selected hyperparameter. This
indicates to me that the method is unreliable. Moreover, if for example
the results in the other figures were based on selecting the max results
across such a sweep, then we would expect a maximization bias leading to
unreliable results.

**Essential References Not Discussed:**

None.

**Experimental Designs Or Analyses:**

Yes, I checked them.
See the Claims and Evidence section for details for the issues.
There are issues with too few seeds used in the experiments and
missing ablations.

**Methods And Evaluation Criteria:**

Yes, the methods and evaluation criteria make sense. The choice of tasks
to tackle is a strong point of this paper as it shows interesting applications
of reinforcement learning.

**Other Comments Or Suggestions:**

Grammatical errors:
Page 1: "hinders" -> "hinder"

Around Line 301: "records" -> "record"

Page 6: "simulates" -> "simulate"

"To further visualize the consistency of between the RM
and the agent, we calculate the errors between RM estima-
tions and precise evaluation during the training process."
Fix the grammar.

Equation 1 for the objective seems incorrect, as it does not take into
account the randomness (you define transition probabilities P, so it
seems you assume randomness in the dynamics).

Equations 5 and 6 are not PPO, they are REINFORCE.


Suggestions:

If you look at Figure 7, it seems that as the training progresses,
in the middle, the model becomes worse at predicting the early rewards.
Perhaps this could be fixed by training the reward model on the stored
rewards from the early stage as well? The model predictive performance could
probably be made to not drop much on the early data.

**Other Strengths And Weaknesses:**

Other strengths:
The ideas are reasonable from a practical point of view.
I liked the choice of tasks.

Other weaknesses:
The clarity could be improved in some places, and there are some
grammatical errors.

Regarding clarity, a few examples are: when you first introduce synchronous
correction, it's not clear what the correction term is (one has to read
the experimental sections to understand, and it may be better to add early
pointers or keep it self-contained), the number of seeds is not clear.

**Questions For Authors:**

How many seeds did you use for all experiments?

**Relation To Broader Scientific Literature:**

The idea of creating a reward model for tasks where the reward evaluation
is slow is a simple one, and it exists in prior work as well. What I found
interesting in the current work was the choice of tasks. The approach
itself is a sensible engineering solution to the tasks that they consider.

**Theoretical Claims:**

No theoretical claims.

---

> ### Author Rebuttal · Authors · 2025-04-01
>
> **Q1:** Experiments with more seeds should be reported.
>
> **A1:** In our submission, we reported results based on 3 seeds. To address your concern, we have now conducted experiments with **10 seeds**. The results (see table in comments for details) confirm that our method remains **robust across a larger number of seeds** and consistently **outperforms the SOTA baselines** across all evaluated scenarios. Notably, the improvements is more evident with 10 seeds, as the reduced standard deviations indicate greater statistical reliability.
>
> We will update the results in Section 5 to further validates the effectiveness and robustness of our approach.
>
> **Q2:** More ablation study.
>
> **A2:** We have conducted additional ablation studies on the module described in Section 4.4 and the results are as follows:
>
> | Full | w/o Synchronous Correction | w/o Model Warm-up | w/o Parallel Computation |
> |-|-|-|-|
> |10.5$\pm$ 0.6 \| 0.31$\pm$ 0.06 | 10.1$\pm$ 0.8 \| 0.29$\pm$ 0.07 | 9.8$\pm$ 0.2 \| 0.24$\pm$ 0.04 | 9.5$\pm$ 0.5 \| 0.29$\pm$ 0.03 |
>
> Removing any component results in a performance decline. Notably, excluding **model warm-up** leads to random rewards in the initial iterations, causing suboptimal policy optimization and a 7% drop in performance. Similarly, removing **parallel computation** reduces training efficiency by limiting the reward model’s access to sufficient samples for accurate fine-tuning, resulting in an approximate 10% performance decrease. These results show that each component contributes to the final performance. We will include these results into a new ablation study section (Section 5.6) of the revised version.
>
> **Q3:** Only provides result for the selected hyperparameter, which is unreliable.
>
> **A3:** Besides the grid search results in Figure 8, we would like to provide **practical guidelines** that enable users to efficiently estimate an **approximate range for the optimal hyperparameters**, ensuring the method’s reliability and reproducibility without exhaustive tuning.
>
> First, the optimal number of **fine-tuning interval** can be approximated by the complexity of the reward function, which is **positively correlated with its evaluation time**. For example, Table 2 indicates that reward computation in pandemic control is **more time-consuming** than molecular design, and grid search identified an optimal fine-tuning interval of 9 and 7 (Figure 8a) for the two scenarios, respectively, suggesting that a more complex reward function requires additional samples for effective RM fine-tuning.
>
> Second, the optimal number of **fine-tuning epoch** should balance the fine-tuning duration with the policy optimization (see Section 5.5). Specifically, we can measure the time $t_1$ required for one sampling and policy optimization iteration. Then, based on the previously determined fine-tuning interval and the reward function’s computation time, we can estimate the number of samples needed for fine-tuning and compute the time $t_2$ per fine-tuning epoch. **The theoretically optimal number of epochs is $N \approx \frac {t_1}{t_2}$**, providing a robust starting point. Search within a small neighborhood of this estimated value typically yields the optimal setting, as validated in our experiments.
>
> We will add the above guidelines on hyper-parameters as Section 4.5 of the revised version.
>
> **Q4:** The RM (reward model) at late-stage becomes worse at predicting the early rewards, using early reward to fine-tune may help.
>
> **A4:** Thanks for your suggestion. We would like to first clarify that the reward prediction error you have observed **does not undermine the performance of our method**, as the RM’s role is to provide accurate evaluations for the current policy rather than historical ones. Moreover, it is worth emphasizing that Figure 7 showcases our AdaReMo enables the RM to adapt to the current policy’s outputs, as evidenced by **near-zero error values along the diagonal elements**.
>
> To further address this, we include samples from previous fine-tuning cycles into RM updates. The updated results for Figure 7 are shown below:
>
> | RM\Precise Reward | 50 | 100 | 150 | 200 |
> |-|-|-|-|-|
> | 50|**0.87%/1.92%**|3.81%/4.47%|8.13%/5.67%|13.16%/10.41%|
> | 100|9.47%/9.12%|**1.32%/1.59%**|8.61%/3.55%|26.78%/16.71%|
> | 150|12.35%/8.21%|4.13%/5.19%|**1.31%/2.10%**|7.25%/6.33%|
> | 200|4.21%/7.92%|5.04%/6.30%|2.74%/1.52%|**0.04%/0.38%**|
>
> The first value uses only current-cycle samples; the second includes previous-cycle samples. Though off-diagonal errors decreased after using historical samples, **diagonal errors increased significantly**, reducing the RM’s ability to accurately evaluate the current policy and **resulting in a 5% performance drop**. This confirms that focusing RM updates on current samples enhances overall optimization effectiveness.
> We have included the above experiments into Section 5.4 of the revised manuscript.
>
> **Q5:** Equation expressions.
>
> **A5:** Please see our response to Reviewer pywE’s Q3.

---

> > ### Comment · Reviewer_kWsg · 2025-04-04
> >
> > Thank you for the rebuttal.
> >
> > >Q3: Only provides result for the selected hyperparameter, which is unreliable.
> >
> > >grid search identified an optimal fine-tuning interval of 9 and 7 (Figure 8a) for the two scenarios
> >
> > These sections seemed to misunderstand my comment, sorry for any confusion.
> > If you look at Figure 8, where you tested the number of fine-tuning intervals in [1, 3, 5, 7, 9]) and the number of epochs  in [10, 20, 30, 40, 50], the performance is only good for the choice (7, 40). All other choices give erratic poor performance. This seems very unreliable to me; it shouldn't be that sensitive. This is a major concern for me, and I did not see how the rebuttal fixes this concern.
> >
> > Moreover, in your rebuttal you mentioned experiments with 10 seeds, but I don't see which Tables you meant.
> >
> > Another remaining question I have is regarding the synchronous correction. While you did address the question and added ablations, I was also interested in the result when only the $r_c$ term is used i.e. $\alpha=0$. (sorry if this was no clear in my review.)

---

> > > ### Author Response · Authors · 2025-04-05
> > >
> > > Dear Reviewer kWsg,
> > >
> > > Thank you so much for your prompt reply! We truly appreciate your continued engagement and constructive feedback. We would like to provide the following response to address your remaining concerns.
> > >
> > > ---
> > > **Q1:** Hyperparameter sensitivity.
> > >
> > > **A1:** We sincerely apologize for misunderstanding your earlier comments on this topic. We now provide a thorough analysis with new experimental results to show that **our method is reliable and achieves stable performance across a wide range of hyperparameter values.**
> > >
> > > **First**, we would like to clarify that the step sizes of grid search in Figure 8 (2 for fine-tuning interval and 10 for the number of epochs) correspond to significant changes in the number of fine-tuning samples. For example, increasing the fine-tuning interval by 2 can introduce up to 150 additional samples, which has a substantial impact on performance. Therefore, we have conducted experiments with fine-grained values for fine-tuning interval (6, 7, 8, 9) and the number of epochs (35, 40, 45, 50):
> > >
> > > |Interval|6|7|8|9|
> > > |-|-|-|-|-|
> > > |MG(FA7)|10.32 (-1.3%)|**10.46**|10.21 (-2.4%)|10.06 (-3.8%)|
> > > |PC(CA-GrQc)|37.52 (-4.7%)|**39.37**|38.19 (-3.0%)|37.94 (-3.6%)|
> > >
> > > |Epoch|35|40|45|50|
> > > |-|-|-|-|-|
> > > |MG(FA7)|10.02 (-4.2%)|**10.46**|10.41 (-0.4%)|9.99 (-4.5%)|
> > > |PC(CA-GrQc)|37.41 (-5.0%)|**39.37**|39.09 (-0.7%)|38.66 (-1.8%)|
> > >
> > > We can observe that **performance remains within 5.0% of the optimal value across different hyperparameter values, showing the reliability of our method.**
> > >
> > >
> > >
> > > **Second**, results in Figure 8 were abtained after 300 optimization iterations. To further assess sensitivity, we extended optimization iterations to 500 and found that **all tested hyperparameters eventually converge to near-optimal performance, albeit with varying training efficiency, as shown in the table below.**
> > >
> > >
> > > | Interval | 5 | 6 | 7 | 8 | 9 | 11 |
> > > |-|-|-|-|-|-|-|
> > > |GA(FA7)| 10.04 (-4.0%) | 10.32 (-1.3%) | **10.46** | 10.33 (-1.2%) | 10.37 (-0.9%) | 10.24 (-2.1%) |
> > > |Best Iter| 409 | 312 | **272** | 296 | 387 | 449 |
> > > |PC(CA-GrQc)  | 38.10 (-3.2%) | 38.40 (-2.5%) | **39.37** | 38.95 (-1.1%) | 38.32 (-2.7%) | 38.11 (-3.2%) |
> > > |Best Iter| 447 | 352 | **216** | 322 | 343 | 463 |
> > >
> > > |Epoch | 30 | 35 | 40 | 45 | 50 | 60 |
> > > |-|-|-|-|-|-|-|
> > > |GA(FA7)| 10.10 (-3.4%) | 10.34 (-1.1%) | **10.46** | 10.42 (-0.4%) | 10.38 (-0.7%) | 10.17 (-2.8%) |
> > > |Best Iter| 461 | 335 | **272** | 310 | 394 | 471 |
> > > |PC(CA-GrQc)| 37.90 (-3.7%) | 38.98 (-1.0%) | **39.37** | 39.09 (-0.7%) | 38.85 (-1.3%) | 38.51 (-2.2%) |
> > > |Best Iter| 380 | 354 | **272** | 292 | 382 | 433 |
> > >
> > >
> > > **Our method consistently converges to high-quality solutions with less than 4% differences to the optimal one, under any hyperparameters of finetuning intervals from 5 to 11 and number of epochs from 30 to 60, though they may cost longer time to converge as reflected by Best Iter in the table.**
> > >
> > > Thank you again for raising this important point. To illustrate that our approach is not highly sensitive to hyperparameters, we have added the above discussions to Section 5.5 of the revised manuscript.
> > >
> > > ---
> > > **Q2:** Results for 10 seeds
> > >
> > > **A2:** We are so sorry for the omission of the table. Below, we present the performance of both the SOTA method and our approach under 3 and 10 random seeds to show the robustness of our method.
> > >
> > > | Seeds | Method | MG(FA7) T5↑\|HR↑ | PC(CA-GrQc) H↑\|C↑ | SP(HLG) D↓\|G↑ |
> > > |-|-|-|-|-|
> > > |3|SOTA|10.3±0.5\|0.25±0.05|36.7±5.2\|9.9±2.4| 3.06±0.21\|2.60±0.13|
> > > |3|Ours|**10.5±0.6\|0.31±0.06**|**39.4±5.7\|10.6±2.9**|**2.88±0.23\|2.80±0.42**|
> > > |10|SOTA|10.1±0.2\|0.22±0.04|35.8±2.3\|9.8±1.4| 3.01±0.11\|2.63±0.07|
> > > |10|Ours|**10.4±0.2\|0.29±0.03**|**39.8±3.4\|10.4±1.2**|**2.82±0.18\|2.84±0.24**|
> > >
> > > **Our method consistently outperforms SOTA baselines across all scenarios under 10 seeds, which reinforces the statistical reliability of the results.** We have incorporated these findings into the revised paper.
> > >
> > > ---
> > > **Q3:** Synchronous correction with $\alpha=0$.
> > >
> > > **A3:** Thank you for highlighting this point. We have conducted additional experiments with $\alpha = 0$. The results are summarized below:
> > >
> > > |$\alpha$|MG(FA7) T5↑\|HR↑|PC(CA-GrQc) H↑\|C↑|SP(HLG) D↓\|G↑|
> > > |-|-|-|-|
> > > |0.8(default)|10.5\|0.31|39.4\|10.6|2.88\|2.80|
> > > |0|9.9 (-5.7%)\|0.17 (-45.2%)|33.9 (-14.0%)\|8.5 (-19.8%)|3.12 (+8.3%)\|2.67 (-7.3%)|
> > >
> > > Setting $\alpha=0$ **introduces greater stochasticity and bias, leading to an average performance drop of about 17%, with a maximal degradation of up to 45.2%**. In some cases, it even underperforms models trained with heuristic metrics. We have included this analysis in the revised manuscript.
> > >
> > > ---
> > > Thank you once again for your follow-up and thoughtful questions. We sincerely hope that our responses have addressed your concerns and highlighted the strengths and contributions of our work.
> > >
> > > If you find our responses satisfactory, we would greatly appreciate your consideration in raising your score.
> > >
> > >
> > > Sincerely,
> > >
> > > All authors

---

### Official Review · Reviewer_pywE · 2025-03-18

**Overall Recommendation:** 2

**Summary:**

The paper proposes **AdaReMo (Adaptive Reward Modeling)**, a reinforcement learning (RL) framework designed specifically for **expensive-to-evaluate reward systems**, such as molecular generation, epidemic control, and spatial planning. The key innovation is the adaptive decomposition of complex, computationally expensive reward functions into localized, neural network-based reward models (RMs). AdaReMo dynamically updates these RMs by fine-tuning them on recent agent trajectories, thus ensuring accurate reward estimations aligned with the agent’s evolving policy. This approach effectively separates fast policy decisions from slow, costly reward evaluations by handling them asynchronously through online and offline systems. Empirical results demonstrate that AdaReMo achieves significant speedups (over 1,000x faster than baselines) and performance improvements (around 14.6%) across the tested tasks, confirming its efficacy in addressing the computational bottleneck present in RL for real-world systems.

**Claims And Evidence:**

Training reinforcement learning (RL) agents requires extensive trials and errors, which becomes prohibitively time-consuming in systems with costly reward evaluations.

The major claim that this paper makes is that the proposed adaptive reward modeling (AdaReMo) paradigm surpasses existing methods in significantly reducing computational overhead and optimality.

I am not fully convinced by this claim since there are several RL optimization algorithms that serve as substitutes for the proposed AdaReMo as well as the compared baselines in the paper. Those RL algorithms include REINFORCE/REINFORCE++, RLOO, GRPO, that circumvents the need for a critic model and brings efficiency.

**Essential References Not Discussed:**

N/A

**Experimental Designs Or Analyses:**

I checked all the experimental designs and analyses. The major concern I have is for including other RL optimization algorithms for comparison.

**Methods And Evaluation Criteria:**

In Eq (7), all the node embeddings are averaged and sent to the MLP layer to calculate the reward, which is inconsistent with Eq (4) where each edge is regarded as an action.

**Other Comments Or Suggestions:**

Typo:

In Eq (7), the h and a should be flipped?

**Other Strengths And Weaknesses:**

[+] The paper is well organized and carefully written, making it quite easy to follow.
[+] The design seems overall reasonable to me. Detaching the reward model to online split and offline split should improve the efficiency of the RL online learning process.

[-] The motivation is still not quite clear to me. If the reward model is of the same size as the policy model (same as the case in RLHF for LLMs), then it is acceptable to obtain the reward. In this paper, the GNN policy and reward model seems even more tiny than most of the open-source LLMs. It is not clear to me why detach the reward modeling to online/offline splits.

**Questions For Authors:**

See other sections.

**Relation To Broader Scientific Literature:**

N/A

**Theoretical Claims:**

I did check all the equations.

---

> ### Author Rebuttal · Authors · 2025-04-01
>
> Dear Reviewer pywE,
>
> We express our sincere gratitude for your thorough review and valuable feedback on our paper. Regarding the potential alternative algorithms and the issue of online and offline splits you mentioned, we hope the following replies can address your concerns.
>
>
> **Q1:** REINFORCE/REINFORCE++, RLOO and GRPO serve as substitutes for the proposed AdaReMo.
>
> **A1:** Thanks for your constructive comments. We would like to provide more clarification on the necessity of the proposed AdaReMo compared to substitutes like RLOO and GRPO with additional experiments.
> In fact, although the introduction of the reward model in AdaReMo draws inspiration from RLHF, its primary focus diverges from the RL optimization algorithms mentioned, such as REINFORCE, RLOO, and GRPO. For instance, GRPO avoids using a critic model to compute advantages due to the typically large size of value networks in RLHF. In contrast, the value network in AdaReMo is significantly smaller, rendering the removal of the critic model less critical. Instead, **AdaReMo targets the efficiency imbalance caused by a computationally expensive reward function**, which sets it apart from these alternatives.
>
> To evaluate the optimization efficiency of various algorithms, we conducted experiments by replacing PPO with RLOO and GRPO. The results are summarized in the table below:
>
>
> ||Molecular Generation(FA7)|Pandemic Control(CA-GrQc)|Urban Spatial Planning(HLG)
> |-|-|-|-|
> ||T5$\uparrow$ \| HR$\uparrow$|H$\uparrow$ \| C$\uparrow$|D$\downarrow$ \| G$\uparrow$
> | RLOO | 10.2$\pm$ 0.4 \| 0.25$\pm$ 0.02 | 34.6$\pm$ 3.8 \| 9.9$\pm$ 1.5 | 3.66$\pm$ 0.38 \| 2.56$\pm$ 0.36|
> | GRPO | 9.6$\pm$ 0.2 \| 0.27$\pm$ 0.03|33.1$\pm$ 4.7 \| 8.7$\pm$ 2.1|3.31$\pm$ 0.24 \| 2.28$\pm$ 0.39|
> |PPO| **10.5**$\pm$ 0.6 \| **0.31**$\pm$ 0.06 | **39.4**$\pm$ 5.7 \| **10.6**$\pm$ 2.9 | **2.88**$\pm$ 0.23 \| **2.80**$\pm$ 0.42 |
>
>
> As demonstrated, **AdaReMo combined with PPO consistently outperforms RLOO and GRPO** across multiple metrics and domains. While RLOO and GRPO eliminate the need for a critic model, **they require multiple sampling**, which increases computational complexity and hinders training efficiency. Furthermore, **the training time per iteration for RLOO and GRPO does not significantly differ from that of PPO**, suggesting that optimizing the critic model is not the primary bottleneck in training efficiency. Additionally, GRPO’s method of calculating advantages **can amplify minor differences**, potentially destabilizing the policy optimization process.
>
> In summary, these results validate the effectiveness of AdaReMo in addressing reward function complexity. Thank you again for your valuable feedback, and we will incorporate the above discussion and experiments into the appendix of the revised manuscript.
>
>
> **Q2:** Why detach the reward modeling to online/offline splits.
>
> **A2:** Thank you for your comment. The online/offline splits are proposed to **address the computationally expensive reward functions**, rather than reducing the computational cost of solution generation. Specifically, **the online system** leverages the reward model to deliver rapid reward estimations, enabling the agent to receive **real-time feedback**. Without this mechanism, the agent would face significant slowdowns due to the time-intensive reward calculations, as exemplified by applications like RLGN (Meirom et al., 2021) for pandemic control. Conversely, **the offline system** continuously refines the reward model using the latest exploratory samples **to ensure its accuracy**. In the absence of this offline phase, the reward model would fail to provide reliable evaluations for samples generated by the evolving policy, leading to significant errors throughout the optimization process, as illustrated in Figure 6a. In summary, the online/offline splits **effectively balance real-time efficiency with long-term accuracy**.
>
> Thank you again for your valuable comment, and we will include the above discussion in Sections 4.3 and 5.4 of the revised manuscript.
>
>
> **Q3:** Typo in Eq4 and Eq7.
>
> **A3:** Thanks for pointing out this issue. We have corrected it in the revised manuscript. Specifically, Eq4 should be $s_i = \text{MLP}_p(\bf{a}_i)$, where $\bf{a}_i$ denotes the embeddings of action ${a}_i$.
> We have checked the entire paper and added following corrections as Reviewer kWsg suggested,
> - Eq1. $\max_{\Theta} E_{\pi_\Theta} \left[ \sum_{t=0}^{T} \gamma^t r(s_t, a_t) \right]$
> - Eq5. $\nabla_{\Theta} J(\Theta) = E \left[ \min \left( r_t(\pi_\Theta) \hat A_t, \text{clip}(r_t(\pi_\Theta), 1 - \epsilon, 1 + \epsilon) \hat A_t \right) \right]$
>
> We hope these responses fully address your concerns and welcome any further questions or suggestions you may have.

---

### Official Review · Reviewer_Q9Xy · 2025-03-26

**Overall Recommendation:** 4

**Summary:**

The authors propose **Adaptive Reward Modelling (AdaReMo)**, an approach that accelerates Reinforcement Learning (RL) by fine-tuning a reward model (RM) multiple times during training. This reduces the need for expensive reward evaluations. In AdaReMo, the RL agent interacts only with the learned RM, while a parallel process computes ground-truth rewards (which are expensive to obtain) to gradually build a fine-tuning pool. Once this pool reaches a threshold (after a fixed number of RL iterations), the RM is fine-tuned for a pre-defined number of epochs and then redeployed in the RL loop. To ensure reliability, the RM is pre-trained for several epochs before any policy optimization begins. PPO is used as the underlying RL algorithm.

Through evaluations on tasks from **Molecular Generation**, **Epidemic Control**, and **Urban Spatial Planning**, the paper shows that AdaReMo outperforms classical,  model-free and model-based RL baselines.

**Claims And Evidence:**

The claims about the effectiveness of AdaReMo are well-supported by the empirical results.

**Essential References Not Discussed:**

None that are obviously missing.

**Experimental Designs Or Analyses:**

The experimental design is clearly described and appears sound.

**Methods And Evaluation Criteria:**

The methodology and evaluation criteria are appropriate and well-justified.

**Other Comments Or Suggestions:**

- Line 235: "reduce-scaled direct evaluations" → *reduced-scale direct evaluations*
- Line 301: "records" → *record*
- Line 313: "have" → *has*
- Line 319: "KDE" → *KED*
- Line 404: "molecular" → *molecules*

**Other Strengths And Weaknesses:**

**Strengths**:
- Clear, concise writing.
- The algorithm design is sensible and addresses a practical bottleneck in real-world RL settings.

**Weaknesses**:
- No major weaknesses observed.

**Questions For Authors:**

1. **Is the warm-up cost included in the plots in Figure 6a?**
   Clarifying this would help interpret the reported efficiency gains of AdaReMo compared to the baselines. If it is excluded, a brief discussion of the warm-up cost tradeoff might be valuable.

**Relation To Broader Scientific Literature:**

**Key Contribution**:
- A method for *adaptive and efficient reward modelling* that avoids expensive reward evaluations by strategically fine-tuning a reward model.

This aligns with ongoing work in sample-efficient RL and learned reward estimation, while targeting real-world domains with costly simulation steps.

**Theoretical Claims:**

No formal theoretical results are presented, but the approach is grounded in sound reasoning.

---

> ### Author Rebuttal · Authors · 2025-04-01
>
> Dear Reviewer Q9Xy,
>
> We would like to express our sincere gratitude for your thorough review and constructive feedback on our paper. We are particularly pleased that you have recognized the effectiveness of our proposed approach, AdaReMo, in solving the efficiency bottleneck between policy optimization and computationally expensive reward calculation. We hope the following responses address your concerns.
>
> **Q1:** Is the warm-up cost included in Figure 6a? A brief discussion of the warm-up cost tradeoff might be valuable.
>
> **A1:** Thank you for your insightful comments. Figure 6a illustrates the training process from when policy optimization begins, and the warm-up cost is not included. Specifically, the model warm-up takes only one fine-tuning interval, accounting for only **3% of the total optimization time**. We also conducted additional ablation experiments by removing the warm-up phase. Without model warm-up, we observed a performance **decrease of 7% across the evaluated metrics**. This decline arises because the policy is initially trained with randomly generated rewards, which introduces additional noise and hinders optimization. These results underscore the critical role of model warm-up in enhancing training efficiency, leveraging a modest (3%) time investment to yield substantial improvements in model performance. We will add the above results and further discussion into a new ablation study section (Section 5.6) of the revised version.
>
>
> **Q2:** Typos and grammatical errors.
>
> **A2:** We sincerely thank the reviewer for their thorough and careful scrutiny. In response to your feedback, we have thoroughly reviewed the manuscript, correcting all identified typos and grammatical mistakes.
>
>
> We hope the above replies resolve your concerns and welcome any further questions.

---

### Decision · Program_Chairs · 2025-05-01

**Decision:**

Accept (poster)

**Comment:**

This submission proposes AdaReMo, a framework for reinforcement learning in domains where reward evaluations are computationally expensive. The key contribution is a modular design that decouples the slow reward evaluation process from the faster policy optimization loop via adaptive, fine-tuned reward modeling. The authors demonstrate the effectiveness of their approach across diverse and challenging domains—molecular generation, epidemic control, and spatial planning—with substantial empirical gains in both training efficiency and policy performance.

The reviews were overall positive. Three reviewers (Q9Xy, kWsg, and joeE) recommended acceptance, citing the paper's well-motivated problem, clear design, and promising empirical results. Reviewer pywE initially expressed concerns regarding baseline comparisons and the motivation for splitting the reward system into online/offline modules. However, the authors’ rebuttal was comprehensive and addressed these concerns convincingly by providing additional experimental baselines (RLOO and GRPO), detailed ablation studies, and fine-grained sensitivity analyses. In addition, Reviewer kWsg subsequently raised their score and acknowledged that the concerns regarding hyperparameter sensitivity had been sufficiently addressed.

While some lingering concerns remain (e.g., training stability and robustness over reward model parameters), the authors have provided detailed empirical evidence (including results from ten random seeds and additional parameter sweeps), which demonstrates robustness and convergence. The learning curves are at times non-monotonic, but the performance at convergence is mostly consistent, and in practice, the best-iteration criterion is often a reasonable standard in domains with expensive evaluations.

This paper provides contributions to the reinforcement learning community. Its modular and adaptive treatment of reward modeling offers a pragmatic solution to a growing class of real-world problems where reward signals are sparse, delayed, or expensive to compute. As a result, I recommend acceptance.